# Amelioration of Congenital Tufting Enteropathy in EpCAM (TROP1)-Deficient Mice via Heterotopic Expression of TROP2 in Intestinal Epithelial Cells

**DOI:** 10.3390/cells9081847

**Published:** 2020-08-06

**Authors:** Gaku Nakato, Sohshi Morimura, Michael Lu, Xu Feng, Chuanjin Wu, Mark C. Udey

**Affiliations:** 1Intestinal Microbiota Project, Intestinal Ecosystem Regulation Group, Kanagawa Institute of Industrial Science and Technology, Kawasaki-shi, Kanagawa 210-0821, Japan; 2Department of Dermatology, International University of Health and Welfare, Narita-shi, Chiba 286-8520, Japan; morimuras-der@iuhw.ac.jp; 3Experimental Immunology Branch, National Cancer Institute, Bethesda, MD 20892, USA; lumi@mail.nih.gov; 4Retired from National Cancer Institute, Bethesda, MD 20892, USA; xuf99@yahoo.com; 5Laboratory of Immune Cell Biology, National Cancer Institute, Bethesda, MD 20892, USA; wuchu@mail.nih.gov; 6Dermatology Division, Department of Medicine, Washington University School of Medicine, Saint Louis, MO 63110, USA; udey@wustl.edu

**Keywords:** TROP1(EpCAM), TROP2, CTE, intestinal epithelial cells

## Abstract

TROP1 (EpCAM) and TROP2 are homologous cell surface proteins that are widely expressed, and often co-expressed, in developing and adult epithelia. Various functions have been ascribed to EpCAM and TROP2, but responsible mechanisms are incompletely characterized and functional equivalence has not been examined. Adult intestinal epithelial cells (IEC) express high levels of EpCAM, while TROP2 is not expressed. EpCAM deficiency causes congenital tufting enteropathy (CTE) in humans and a corresponding lethal condition in mice. We expressed TROP2 and EpCAM in the IEC of EpCAM-deficient mice utilizing a villin promoter to assess EpCAM and TROP2 function. Expression of EpCAM or TROP2 in the IEC of EpCAM knockout mice prevented CTE. TROP2 rescue (T2R) mice were smaller than controls, while EpCAM rescue (EpR) mice were not. Abnormalities were observed in the diameters and histology of T2R small intestine, and Paneth and stem cell markers were decreased. T2R mice also exhibited enlarged mesenteric lymph nodes, enhanced permeability to 4 kDa FITC-dextran and increased sensitivity to detergent-induced colitis, consistent with compromised barrier function. Studies of IEC organoids and spheroids revealed that stem cell function was also compromised in T2R mice. We conclude that EpCAM and TROP2 exhibit functional redundancy, but they are not equivalent.

## 1. Introduction

TROP1 (EpCAM, CD326) and TROP2 are homologous type I cell surface proteins that were initially identified on maternal tissue-invasive placental trophoblasts [1]. It is now recognized that these proteins are widely expressed, and sometimes co-expressed, in developing and adult epithelia in many tissues in humans and other vertebrates, and by epithelial cancer (carcinoma) cells and tissue and embryonic stem cells as well [2,3,4]. There is substantial literature regarding the role of EpCAM and TROP2 in cancer [5,6]. Although there are inconsistencies, on balance it appears that expression of TROP2 or EpCAM typifies a more aggressive, invasive tumor and portends a poor outcome.

TROP2 and EpCAM collectively comprise a 2-gene family, encoding ~35 kDa proteins with ~50% amino acid identity and ~66% similarity [4]. Each protein features a single extracellular region containing thyroglobulin- and EGF-like domains, transmembrane domains that are highly homologous and short cytoplasmic domains. The locations of cysteine residues in the extracellular regions of TROP2 and EpCAM are almost invariant, suggesting that the secondary structures of the extracellular portions of these two molecules are very similar. The cytoplasmic domains of TROP2 and EpCAM are dissimilar, consistent with the concept that engagement of the extracellular domains by as yet unknown ligands could trigger different signaling cascades.

Mechanisms by which TROP2 and EpCAM effect changes in cell behavior are incompletely characterized. Both molecules are subject to regulated intramemenbrane proteolysis (RIP) mediated by the sequential action of TACE and presenilins that results in the generation of soluble intracellular peptides that bind to TCF/LEF transcription factors, thus modifying gene expression in cancer cell lines and conferring aspects of stem cell function [2,5]. Both TROP2 and EpCAM also bind to and stabilize the tight junction-associated proteins claudin-1 and claudin-7, regulating tight junction composition and epithelial barrier function [7].

We sought to test the functional equivalence of TROP2 and EpCAM using a stringent in vivo assay. EpCAM is expressed by intestinal epithelial cells (IEC) of the developing and adult gut, whereas TROP2 is expressed only in the former [8]. Truncating and null mutations of *EpCAM* in humans and mice cause congenital tufting enteropathy (CTE), a severe diarrheal disorder characterized by epithelial dysplasia, compromised intestinal barrier, failure to thrive, and, in mice, post-natal demise within the first week of life [9,10]. CTE is rare disorder [11] and the underlying molecular pathogenesis of CTE remains unknown. To examine the potential function redundancy of these two molecules in CTE, we expressed transgenes encoding murine TROP2 (mTROP) or human EpCAM (hEpCAM) in the IEC of C57BL/6 mice using a villin promoter and assessed the ability of each transgene to ameliorate CTE in *EpCAM^−/−^* mice. Our results indicate that TROP2 can prevent the development of symptomatic CTE in *EpCAM^−/−^* mice and, also, that TROP2 and EpCAM are not equivalent.

## 2. Materials and Methods

### 2.1. Animals

C57BL/6 mice were purchased from the NCI Frederick National Laboratory (Frederick, MD, USA). Mice were bred and maintained in a pathogen-free environment. Experiments involving animals were approved by the NCI Animal Care and Use Committee (DB-054 and DB-054 M7).

### 2.2. Generation of Transgenic Founder Mice Expressing Murine TROP2 and Human EpCAM in Murine IEC

Transgenic mice expressing murine TROP2 and human EpCAM in IEC were generated using a villin promotor [12] and full-length murine TROP2 and human EpCAM cDNA prepared from corresponding pCMV6 backbone expression plasmids (Origene, Rockville, MD, USA). Relevant PCR primers are described in Appendix A. Purified cDNA fragments were inserted into Xho I/Cla I cloning sites of 12.4 kb villin-ΔATG (Addgene, Watertown, MA, USA). Pme I was used to release villin-murineTROP2 or villin-human EpCAM from vector before the injection into zygotes. Transgenic mice were generated at the NCI CCR Transgenic Mouse Model Laboratory. Transgenic mice carrying villin-mTrop2 or villin-hEpCam were identified by PCR genotyping of tail DNA (see Appendix A) using specific primer sets (Appendix A).

### 2.3. Generation of EpCAM Germline Deletion Knock-Out Mice Rescued by Murine TROP2 and Human EpCAM Transgene

The conditional *EpCAM^fl/fl^* mouse described previously [13] was crossed to an EIIA cre deleter (B6.FVB-Tg (EIIa-cre) C5379Lmgd/J: Jackson Laboratories) to attain a germline EpCAM null allele in the heterozygous state. The EIIA cre gene was crossed out in a subsequent generation following Jax’s PCR protocol. This *EpCAM^+/−^* founder mouse was then crossed to either the aforementioned *villin-mTROP2* or the *villin-hEpCAM* transgenic founder mouse. An intra- or self-cross of the *EpCAM^+/−^*, *villin-mTROP2^Tg+^* strain or the *EpCAM^+/−^,villin-hEpCAM^Tg+^* strain gave the desired *EpCAM^−/−^ villin-mTROP2 ^Tg+^ or EpCAM^−/−^ hEpCAM ^Tg+^* rescue mice (T2R or hEpR mice, respectively).

### 2.4. Acute EpCAM Silencing

To obtain acute conditional EpCAM knockout mice, we crossed *ROSA26 Cre^ERT2+^* mice (Jackson Laboratory, Bar Harbor, ME, USA) with *EpCAM^fl/fl^* mice that had been produced in our laboratory [13] and treated them with tamoxifen. Tamoxifen (Sigma-Aldrich, St Louis, MO, USA) was dissolved in sunflower oil (33 mg/mL) with sonication (Fisher Scientific, Pittsburgh, PA, USA) and administrated via gavage (0.2 mg/g body weight) to 8–10-week-old *ROSA26 Cre^ERT2+^ EpCAM^fl/fl^* mice daily for 3 days. Intestinal tissues were harvested on day 7.

### 2.5. Quantitation of muTROP2 and huEpCAM Expression via qPCR.

Total RNA was prepared from small intestines using RNeasy Plus Universal Mini Kits (Qiagen, Germantown, MD, USA) and cDNA was synthesized with SuperScript III First-Strand Synthesis SuperMix. Quantitative PCR was performed using Maxima SYBR Green qPCR Master Mix (ThermoFisher Scientific, Carlsbad, CA, USA) and a C1000 Thermal Cycler (BioRad, Hercules, CA, USA). All qPCR primers were obtained from BioRad (PrimePCR SYBR Green Assay). Plasmids encoding murine EpCAM, human EpCAM and murine TROP2 were obtained from Origene. Ten-fold serial dilutions of the known amounts of plasmid DNA, ranging from 1 × 10^4^ to 1 × 10^8^ plasmids/µL, were used to create standard curves for each PCR product and numbers of mRNA molecules/ug total RNA were calculated [14].

### 2.6. Hematoxylin and Eosin and Alcian-Blue-PAS Staining

Intestinal tissues were fixed with 10% formalin for 48 h. Tissues were then rinsed with PBS, stored in 70% EtOH at 4 °C, and sent to Histoserv, Inc (Germantown, MD, USA). for paraffin embedding, sectioning, and staining.

### 2.7. Immunofluorescence Microscopy

Intestinal tissues were fixed with 4% paraformaldehyde for 48 h, rinsed and exposed to increasing concentrations of sucrose (10–30%) at 4 °C and embedded in an Optimal Cutting Temperature (OCT) compound prior to freezing. In selected experiments, fresh tissue was placed directly into OCT. Ten μm-thick frozen sections from pre-fixed tissues were prepared and stained directly with primary antibodies. Flash frozen sections were fixed briefly with cold methanol-acetone to allow detection of claudins. Tissue-bound primary antibodies were detected with Alexa Fluor-conjugated secondary antibodies (ThermoFisher Scientific). As indicated, sections were additionally stained with Alexa Fluor 647 Phalloidin (Cell Signaling Technology, Danvers, MA, USA). Stained sections were mounted on glass slides in ProLong Gold anti-fade regent with a 4′,6-diamidino-2-phenylindole (DAPI) counterstain (Invitrogen, Waltham, MA, USA) and confocal fluorescence microscopic images were obtained using an Axio A1 (Zeiss, White Plains, NY, USA) and Axio Vision Rel. 4.8 software.

### 2.8. Antibodies

The following commercial monoclonal or polyclonal antibodies were utilized at appropriate concentrations: Anti-human/mouse EpCAM C-terminus (ThermoFisher Scientific); anti-Ki67 (abcam, Cambridge, MA, USA); anti-lysozyme (Dako, Via Real Carpinteria, CA, USA); anti-claudin-1 (Invitrogen); anti-claudin-3 (Invitrogen); anti-claudin-7 (Invitrogen); anti-claudin-15 (Invitrogen); anti-HA tag (Sigma, St Louis, MO, USA); anti-mouse β-Actin (AC-15, Sigma); anti-mouse claudin-1 (2H10D10, ThermoFisher Scientific); anti-FLAG tag (M2, Sigma). Several rabbit monoclonal antibodies (anti-murine EpCAM (clone E73) and anti-TROP2 (clone E69)) were generated by Epitomics, Inc. (Burlingame, CA, USA). via a contract.

### 2.9. Electron Microscopy

The intestine was fixed for 2.5 h in 2.5% glutaraldehyde/0.1 M phosphate/0.1 M sodium cacodylate buffer (pH 7.4) and post-fixed in 1% (*v*/*v*) osmium tetroxide in the same buffer for 1 h at room temperature (RT) followed by a graded ethanol dehydration. Ethanol was replaced with tetramethylsilane and tissue was allowed to air dry. Specimens were mounted on SEM studs, sputter-coated with platinum gold, and observed using an S-4500 scanning electron microscope (Hitachi, Dallas, TX, USA). The intestine was fixed for 48 h in 2.5% glutaraldehyde/0.1 M phosphate/0.1 M sodium cacodylate buffer (pH 7.4), cut into 1–2 mm pieces, immersed for 4 h in the same fixative, and post-fixed in 1% (*v*/*v*) osmium tetroxide in the same buffer for 1 h at room temperature, followed by en bloc staining in 0.1% (*w*/*v*) uranyl acetate/0.1 M acetate (pH 4.2) for 1 h. Tissues were dehydrated in graded ethanol and 100% propylene oxide and then infiltrated in an equal volume of propylene oxide and epoxy resin overnight on a rotating table. The tissues were embedded in Epon and cured at 55 °C for 48 h. Ultrathin sections were cut on an ultra-microtome and stained with uranyl acetate and lead citrate prior to observation via a transmission electron microscope H7600 (Hitachi). All electron microscopic experiments were performed by the Electron Microscope Laboratory at Frederick National Laboratory for Cancer Research).

### 2.10. RNA-FISH

Intestinal tissues were fixed in fresh 4% Paraformaldehyde (PFA) for 24 h at RT. Representative 3 mm cross sections were obtained and stored in 70% ethanol (EtOH) at room temperature until embedding in histology grade paraplast (McCormick Scientific). Paraffin-embedded samples were serially sectioned at 5 μm using a rotary microtome (Leica Microsystems, Buffalo Grove, IL, USA). Details of the staining procedure are described in the Appendix A. RNA-FISH was performed by the Molecular Pathology Laboratory at the Frederick National Laboratory for Cancer Research.

### 2.11. Assessment of Intestinal Permeability

Eight–twelve-week-old mice were anesthetized with isoflurane prior to gavage with 50 mg/100 g body weight of fluorescein isothiocyanate (FITC)-dextran (average molecular weight 4000, Sigma) in PBS (80 mg/mL). After 4 h, peripheral blood was collected into BD Microtainers (BD, San Jose, CA, USA) under the isoflurane anesthesia. Serum was isolated from whole blood and fluorescence intensities of each sample were measured using a Victor X3 fluorimeter (Perkin Elmer, Shelton, CT, USA).

### 2.12. Dextran Sulfate Sodium (DSS)-Induced Colitis

DSS (MP Biomedicals, Solon, OH, USA) was added into drinking water (2.5% weight/volume) and administered to mice *ad libitum* for 7 d. Body weights were determined daily and mice were euthanized when they reached predetermined experimental endpoints.

### 2.13. Western Blotting

Intestinal tissues were homogenized in Radioimmunoprecipitation (RIPA) (150 mM NaCl/1.0% Triton X-100/0.5% sodium deoxycholate/0.1% SDS/50 mM Tris; pH 8.0) lysis buffer containing Complete Ultra protease inhibitors (Roche, Indianapolis, IN, USA). Protein concentrations were determined using Pierce BCA protein assay kits (ThermoFisher Scientific) and normalized amounts of protein were denatured and reduced with LDS/DTT buffer (Invitrogen). Five–ten μg of protein was resolved in NuPAGE 4–12% Bis-Tris gels run in a 3-morpholimopropane-1-sulfonic acid (MOPS) buffer (Invitrogen), transferred onto nitrocellulose membranes (Invitrogen), and blocked with 5% non-fat dry milk(NFDM) in Tris-bufferd saline, 0.1% Tween 20 (TBST). EpCAM, TROP2, and claudins were detected with relevant primary antibodies (see above) after overnight incubation at 4 °C. TBST-washed blots were incubated for 1 h at RT with horseradish peroxidase-conjugated secondary Ab (Jackson ImmunoResearch, West Grove, PA, USA), washed, and exposed to SuperSignal West Pico chemiluminescent substrate or SuperSignal West Dura (Thermo Fisher) extended duration substrate. Proteins of interest were detected with X-ray film using a Kodak 2000A film processor. Blots were digitalized using Biorad’s Chemidoc analyzer and Quantity One 4.4.1 software.

### 2.14. Cell Transfection

HEK 293 cells were seeded into 60 mm culture dishes and cultured in Dulbecco’s modified Eagle’s medium supplemented with 10% Fetal calf serum (FCS) at 37 °C with 5% CO_2_. Sequence-verified claudin-1-FLAG PCR fragments, claudin-7-FLAG PCR fragments, and TROP2-HA PCR fragments were cloned into pcDNA3. pcDNA3-HAEpCAM has been described [7]. FLAG-tagged plasmids and HA-tagged plasmids were co-transfected into HEK 293 cells using Lipofectamine 2000 (Life Technologies, Carlsbad, CA, USA).

### 2.15. Immunoprecipitations

Transfected cells were lysed in 150 mM NaCl/1% Triton X-100/1 mM EDTA/30 mM NaF/2 mM sodium pyrophosphate/20 mM Tris (pH 7.5) with 1× protease inhibitor cocktail (Roche). Cell lysates were pre-cleared with Protein G Sepharose 4 Fast Flow for 1 h at 4 °C, incubated with rat anti-HA antibody (3F10, Roche) for 4 h at 4 °C, and then Protein G Sepharose for 1 h. Immunoprecipitates were recovered, washed, resolved in NuPAGE 4–12% Bis-Tris gels (Novex), and immunoblotted with anti-FLAG antibody or anti-HA antibody. Proteins of interest were visualized using HRP-conjugated secondary Ab (Jackson ImmunoResearch) and enhanced SuperSignal West Pico chemiluminescent substrate or SuperSignal West Dura extended duration substrate.

### 2.16. Propagation of IEC In Vitro

IEC organoids were generated from murine intestinal epithelial crypts following the protocol from Clevers et al. [15,16,17]. Briefly, intestinal crypts were isolated from small intestines of WT and Cre^ER^ mice using 5 mM EDTA/PBS with mechanical dissociation. Suspended crypts were passed through 100 and 70 μm nylon mesh and cultured in growth factor-supplemented media. Organoid forming efficiencies were calculated by dividing the total number of organoids at day 9 per well by the total number of organoids at day 1 per well and multiplying by 100. To determine organoid passage efficiencies, organoids were subcultured 1 well into 4 on day 9 of the initial culture period. Organoid passage efficiencies represent the total number of organoids present in all 4 wells on day 9 of the first passage divided by the total number of organoids present in the source well on day 9 of the initial culture period. IEC spheroids were generated from murine intestinal crypts as described by Stappenbeck et al. [18]. Spheroids were studied during the initial culture period and after the fourth passage.

### 2.17. Statistics

Statistical analysis was performed with ANOVA with Tukey’s multiple comparison test or multiple *t*-test with Holm–Sidak methods by PRISM7 software. Differences were considered significant at *p* < 0.05.

## 3. Results

### 3.1. Expression of Murine TROP2 and Human EpCAM in the IEC of Transgenic Mice

We generated transgenic mice expressing murine TROP2 (mTROP2) or human EpCAM (hEpCAM) in the intestinal epithelial cells (IEC) of EpCAM-deficient (*EpCAM^−/−^*) mice utilizing a villin promoter [12] and utilized these animals to test the functional equivalence of EpCAM and TROP2. In contrast to *EpCAM^−/−^* mice that fail to thrive and die within the first week of life, both *EpCAM^−/−^ mTROP2^Tg^* (T2R) and *EpCAM^−/−^ hEpCAM^Tg^* (hEpR) mice were viable and fertile. We confirmed transgene expression in the gastrointestinal tracts of these mice using quantitative PCR (qPCR) and immunofluorescence microscopy (IFM). qPCR demonstrated that the absolute numbers of mTROP2 and hEpCAM transcripts in the intestinal tissue of transgenic mice were higher than endogenous murine EpCAM transcripts in wild type (WT) mice (Figure 1a). IFM results indicated that, like endogenous murine EpCAM, transgenic mTROP2 and hEpCAM were expressed by both villous and crypt epithelial cells (Figure 1b,c). We conclude that hEpCAM and mTROP2 are expressed at relevant levels and in the appropriate places in transgenic mouse intestine, and that we are therefore likely to be able to draw meaningful conclusions from this study.

### 3.2. Transgenic Expression of mTROP2 in the IEC of EpCAM Knockout Mice Prevents Congenital Tufting Enteropathy but does not Restore Normality

Expression of mTROP2 or hEpCAM in the IEC of EpCAM knockout mice prevented CTE that is routinely manifested by diarrhea, failure to thrive, and neonatal demise. Mice of both genotypes steadily gained weight, but T2R mice were smaller than littermate controls (Figure 2a). Significant differences in body sizes were observed from the third to eighth week after birth and after the forty-fourth week as well (Figure 2b). Interestingly, only male T2R mice were smaller than corresponding controls (Appendix A). In contrast, there were no significant differences between the body sizes of male or female hEpCAM rescue mice and their littermate controls (Appendix A).

CTE mouse models recapitulate the epithelial dysplasia with tuft formation and villous atrophy that is seen in patients with this rare disease [19]. We assessed the histology of intestinal tissues from T2R and hEpR mice and compared it with that seen in CTE mice. To reproduce murine CTE, we treated adult *ROSA^CreER^ EpCAM^−/−^* mice with tamoxifen and obtained tissue from mice that had experienced a 20–25% decrease in body weight (AcEp KO mice). Histologic sections from AcEp KO mice revealed typical features of CTE, while sections from T2R and hEpR mice did not (Figure 2c). T2R villous structure was abnormal, however, and the observed abnormalities were most striking in the duodenum. We did not observe histologic abnormalities in the colons of T2R mice (Appendix A). In the small intestine, T2R villi appeared to be longer than in mice of other genotypes, and there was more variation in villous structure in T2R mice than in littermate controls or hEpR mice. Consistent with this, the diameters of samples of duodenum and jejunum from T2R mice were also greater than in controls or hEpR mice (Figure 2d). We additionally characterized villous structure in T2R mice using scanning (Appendix A) and transmission electron microscopy, and did not detect striking differences. Microvilli lengths did not differ between WT, T2R, and hEpR mice (Figure 2e).

### 3.3. Abnormal Intestinal Epithelial Barrier Function in T2R Mice

After observing that mesenteric lymph nodes in T2R mice were larger than those in all other mice (Figure 3a), we hypothesized that the intestinal barrier in T2R mice might be compromised and that this might predispose them to subclinical inflammation induced by microbes or other intestinal constituents. We assessed the intestinal barrier function of the small and large intestines of T2R mice and relevant control mice using a 4 kDa FITC-dextran permeability assay [20] and the dextran sodium sulfate (DSS)-induced colitis model [21]. Quantitation of FITC-dextran concentrations in the sera of mice that had been gavaged with FITC-dextran 4 hrs earlier revealed increased intestinal permeability of T2R mice as compared with WT or hEpR mice (Figure 3b). Exposure of WT, T2R, and hEpR mice to 2.5% DSS in their drinking water for seven days demonstrated that T2R mice were also more susceptible to DSS-induced colitis. Body weights of T2R mice began to decrease on day 1 of exposure to DSS and were statistically different from WT mice by day 4 (Figure 3c). hEpR mice were also somewhat more sensitive to DSS than controls. Body weights of these mice began to decrease on day 4 and they were intermediate between those of WT and T2R mice until day 7 (Figure 3c). Routine histology of colonic tissue from day 5 DSS-treated mice was consistent with the observed changes in body weights. T2R colonic mucosa was heavily infiltrated with lymphocytes and submucosal lymphoid follicles were prominent (Figure 3d). Inflammation of the WT and hEpR colon was much less striking. Goblet cell-derived mucin is an important component of the colonic epithelial barrier. Alcian blue/PAS-staining globlet cell numbers and distributions were similar in the colons of WT, CTL, T2R, and hEpR mice (Appendix A).

### 3.4. Intestinal Epithelial Cell Lineages in T2R Mice

Intestinal epithelial crypt stem cells give rise to specialized Paneth cells, goblet cells, and enteroendocrine cells, in addition to epithelial cells that cover most of the surfaces of villi. Lysozyme^+^ Paneth cells produce antimicrobial peptides and stem cell-sustaining Wnt, goblet cells elaborate PAS^+^ mucin, and enteroendocrine cells secrete peptide hormones.

Expanded intestinal crypts comprising proliferating cells are a feature of CTE [19]. We visualized Ki-67^+^ proliferating cells in histologic sections to gain insights into crypt architecture in AcEp KO, T2R, and control mice. Proliferating cells were clearly increased in the ileum of AcEp KO mice relative to WT and CTL mice as expected (Figure 4a). The zone of proliferating cells in T2R mice was similarly expanded and it appeared to be larger than that present in hEpR mice (Figure 4a). PAS^+^ goblet cells were regularly distributed in the villous epithelium of WT, Ctrl, T2R, and hEpR mice, and they were also identifiable in the disordered epithelia of AcEp KO mice (Figure 4b). Lysozyme^+^ Paneth cells were present in similar numbers and locations in crypts of mice with all genotypes (Figure 4c).

### 3.5. Abnormalities in IEC Stem Cell Function in T2R Mice

The frequency and functional capabilities of IEC stem cells in T2R and relevant control mice was probed using in vivo and in vitro approaches. Stem cells were localized and enumerated in tissue sections using RNA-Scope fluorescence in situ hybridization (FISH) [22] to identify cells containing transcripts encoding the transcription factor Olfm4 [23]. These cells were essentially absent from the intestinal epithelium of AcEp KO mice and were significantly diminished in T2R mice, whereas they were present in similar numbers in WT and hEpR mice (Figure 5a). Assessment of Paneth cells containing transcripts encoding the antimicrobial peptide defensin-1 (Defa1) suggested that these cells might also be less frequent in crypts of T2R mice, but they were clearly more abundant than in AcEp KO mice (Figure 5a).

Additional characterization of stem cells and Paneth cells in these mice was accomplished by quantifying mRNA levels of transcripts that are typical of each cell type using qPCR. These studies revealed that the Olfm4 mRNA contents of T2R intestinal epithelia were much lower than those of WT epithelia, that levels in WT and hEpR mice were similar, and that Olfm4 transcripts were significantly lower in IEC from T2R mice than in WT or hEpR mice (Figure 5b). Assessment of Paneth cell-specific lysozyme mRNA content in T2R and control mice demonstrated that levels were low and similar in AcEp KO and T2R mice as compared with WT mice and that lysozyme transcripts were present at intermediate levels in the intestinal epithelium of hEpR mice (Figure 5c). In aggregate, these results suggest that there is an abnormality in the intestinal epithelial stem cell compartment of T2R mice, and that Paneth cell number and/or function may be abnormal as well. Since stem cell survival and function is influenced by adjacent Paneth cells, it is possible that the abnormalities that we have identified in the stem cell compartment reflect defects in stem cells, Paneth cells, or both types of cells.

To determine if TROP2 was necessary and/or sufficient for stem cell proliferation, we cultured crypt-derived multi-lineage organoids and stem cell-enriched spheroids from the T2R and control mice. T2R organoids were noticeably smaller than control organoids after day 9 of culture (Figure 6a). In addition, T2R organoids had fewer budding domains (composed of Paneth cells and stem cells) than those of the control organoids (Figure 6a). Although fewer crypts were isolated from T2R crypts than other mice, genotype-specific significant differences in organoid forming efficiencies were not observed (Figure 6b). In contrast, the passaging efficiencies of T2R organoids were much lower than those of organoids from WT and littermate control mice (Figure 6b). The ability of hEpR organoids to be passaged was also somewhat impaired. One possibility is that Paneth cell function is marginal in freshly isolated crypts from T2R mice and that it is additionally diluted in organoid cultures.

IEC stem cells can be propagated as spheroids in vitro in a Paneth cell-independent fashion using the recently described culture conditions [18]. We generated spheroids from crypts from WT, Ctrl, T2R, and hEpR mice using this approach. Spheroids from mice of all genotypes grew well initially and after the first passage (Figure 6c). However, with continued passaging, T2R spheroids became smaller (Figure 6c) and they typically were lost prior to the seventh passage. This suggests that the stem cell number and/or function is compromised and perhaps borderline in T2R mice.

### 3.6. Claudin Stabilization in TROP2 Rescue Mice

Previous studies reported that physical interactions between EpCAM and selected claudins prevent the latter from lysosomal degradation [7,10,24]. Claudin-7 (*Cldn7*) binds directly to EpCAM via an AxxxG motif within the transmembrane domain and claudin-1 appears to associate with EpCAM via claudin-7 [2,7]. Moreover, EpCAM loss leads to decreased claudin expression in the intestines of both humans and mice [10,19]. Finally, *Cldn7* knockout mice and *EpCAM* null mice both exhibit a CTE phenotype [25]. We analyzed claudin-1 and claudin-7 expression in T2R mice because the transmembrane domains of TROP2 and EpCAM are highly conserved [4] and we hypothesized that EpCAM and TROP2 would both stabilize claudins in vivo.

We focused on intestinal epithelial claudin protein expression because the claudin transcript levels and protein levels do not necessarily correspond [7]. Western blotting revealed that the amounts of claudin-1 and claudin-7 in the intestinal tissues of T2R mice were lower than in WT or hEpR mice (Figure 7a). We confirmed that claudin-1 and claudin-7 expression was decreased in T2R mice using immunofluorescence microscopy. AcEp KO mice were studied as controls (Figure 7a). As expected, IEC plasma membrane-associated claudin-1 was lost in the villous and crypt regions of AcEp KO mice (Figure 7b). Claudin-1 expression localized to the lateral and apical cell surfaces of IEC of WT, Ctrl, and hEpR mice, but the intensity of claudin-1 staining in hEpR mice was perhaps somewhat lower than in WT and Ctrl mice (Figure 7b). In addition, residual claudin-1 appeared to preferentially accumulate on the apical lateral surfaces of T2R and AcEpKO IEC (Figure 7b). Claudin-7 appeared to be normally distributed in the villous regions of T2R mice, but staining intensity was clearly decreased. In contrast, claudin-7 expression in the crypts of T2R mice was very low and the protein that was present localized to the apical regions of the cells (Figure 7c). Claudin-3 and claudin-15 were also somewhat decreased in T2R mice (Appendix A).

The relative abilities of EpCAM and TROP2 to associate with claudin-1 and claudin-7 were assessed via co-immunoprecipitation studies involving HEK 293 cells and expression plasmids encoding HA-tagged EpCAM and TROP2 and FLAG-tagged claudins. Claudin-7 and claudin-1 each co-immunoprecipitated with TROP2 and EpCAM with similar efficiencies (Appendix A). TROP2 appears to stabilize enough claudin-7 in IEC to circumvent the development of CTE.

## 4. Discussion

TROP2 and EpCAM are homologs that are not closely related to other cell surface proteins [2,4]. The extent to which they complement each other has not been formally addressed. We compared the functional capabilities of TROP2 and EpCAM in vivo in an epithelium in which only 1 of the 2 was normally expressed. Loss of EpCAM expression in IEC resulted in epithelial dysplasia and comprised barrier function that manifested as a severe congenital diarrheal syndrome termed CTE. We utilized a transgenic approach to express either TROP2 or EpCAM in the IEC of *EpCAM^−/−^* mice, and assessed the ability of these transgenes to ameliorate CTE.

TROP2 (T2R) and EpCAM (hEpR) mice both survived, were vigorous, and were fertile. Male (but not female) T2R mice were statistically smaller than littermate controls at some time points. The apparent sex-dependent differences that we observed are unexplained. Serum gluose levels in male T2R mice were lower than those in male control mice, while serum glucose levels in female T2R mice were not different from those in corresponding female control mice. This observation could suggest abnormal nutrient absorption in male T2R mice, but this was not formally tested. Sizes of male and female hEpR mice were not different from those of littermate controls. Alterations in the barrier function of T2R small and large intestines were revealed by assessment of FITC-dextran permeability and sensitivity to DSS-induced colitis, respectively. In this study, we used female mice in the DSS experiments, because male mice tend to fight when non-littermates are co-housed. Male mice tend to be more susceptible to DSS-induced colitis than female mice [26,27]. Taken together, T2R male mice might have revealed more drastic body weight changes compared to female mice, but this has not been tested. Reductions in the frequency and/or function of Paneth cells and intestinal epithelial cells in T2R mice were detected in vivo. CTE patients and mutant mice model had significantly fewer Paneth cells and goblet cells than their healthy counterparts [28]. However, forced expression of murine TROP2 in intestine was sufficient to support adequate Paneth cell numbers for survival in the absence of endogenous EpCAM. There were differences in the distributions of claudin-1 and claudin-7 in T2R IEC as compared to hEpR and wild type mice, but TROP2 and EpCAM associated with claudin-1 and claudin-7 with similar efficiencies when co-expressed in HEK 293 cells. Clinical observation of limited numbers of aged mice did not reveal an obvious cancer predisposition of any genotype, but this question was not evaluated systematically.

In aggregate, our results document that TROP2 and EpCAM are functional as well as structural homologs and that they are not equivalent. Although TROP2 and EpCAM associated with claudin-1 and claudin-7 to similar extents in vitro, their abilities to stabilize claudins in IEC in vivo differed. Decreased levels and altered distributions of claudin-1 and claudin-7 in T2R mice may explain the abnormalities in barrier function that we observed. Disruption of claudin-7 has been shown to alter integrin α2/claudin-1 protein complex distribution [25]. Hence, the accumutration of residual claudin-1 on apical lateral surfaces of T2R and AcEp KO IEC was likely due to decreased claudin-7 expression. Whether or not alterations in claudin metabolism could underlie the stem cell abnormalities that we observed is less certain. The relative decreases in TROP2, claudin-1, and claudin-7 that were observed in crypt IEC as compared with villous IEC coincides with the physical location of Paneth and stem cells. Recently, it has been reported that claudin-7 plays a critical role in stem cell self-renewal and differentiation [29]. Decreases in claudin-7 in T2R might lead to reductions in stem cell numbers. We cannot attribute our findings to low expression of the TROP2 transgene since TROP2 transcript abundance was greater than abundance of endogenous EpCAM transcripts and it was similar to levels of hEpCAM transcripts. Since both mTROP2 and hEpCAM transgenes were driven by the villin promoter, it also seems unlikely that expression of mTROP2 in stem cells was lower than that of hEpCAM.

It is possible that TROP2 is less stable in crypt IEC than EpCAM. We have recently identified and characterized a functional pathway that links HAI-2 (*SPINT2*), matriptase, EpCAM, and claudin-7 in IEC [24]. In the absence of the protease inhibitor HAI-2, the serine protease matriptase cleaves EpCAM, thereby targeting EpCAM and claudin-7 for lysosomal degradation. We have reported that this pathway exists in keratinocytes, and that in these cells, TROP2 is also a matriptase substrate [30]. Perhaps TROP2 and EpCAM turn over at different rates in crypt versus villous IEC via matriptase-mediated or other mechanisms.

Stem cell deficits may be secondary to Paneth cell abnormalities, or be cell autonomous. We probed this question by propagating IEC organoids and spheroids from T2R mice and controls. Differences between T2R mice and controls became evident only after passage, suggesting that stem cell and Paneth cell function may be borderline in crypts in vivo. The observation that T2R spheroids could not be passed in Paneth cell-independent culture suggests that there are defects in stem cells themselves. We cannot exclude the possibility that there are also Paneth cell abnormalities, which could compromise the amounts EGF, Dll4, and Wnt-3 that are available to maintain stem cell niches in T2R mice.

As mentioned earlier, a long literature implicates TROP2 and EpCAM in cancer biology. Both TROP2 and EpCAM are known to associate with claudin-1 and claudin-7. The distinct activities that have been ascribed to these two proteins have, for the most part, been attributed to activation of different signaling pathways that have been linked to differences in the cytoplasmic domains [4]. We propose that both TROP2 and EpCAM can prevent most manifestations of CTE because each can stabilize claudins in IEC, at least to some degree. Theoretically, forced expression of either EpCAM or TROP2 in the intestinal epithelia of CTE patients could also attenuate or ameliorate symptomatology. In addition to the possibility that TROP2 and EpCAM turnover may be differentially regulated in IEC, additional signaling-related activities that are protein-specific may explain why one did not completely substitute for the other in our studies.

## Figures and Tables

**Figure 1 cells-09-01847-f001:**
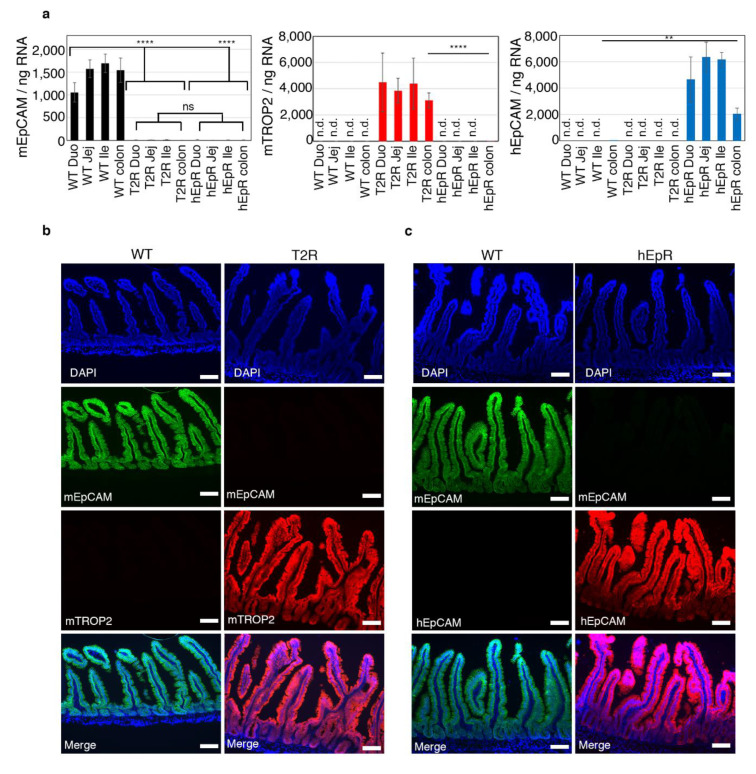
Expression of EpCAM and TROP2 in intestinal epithelial cells (IEC) of transgenic and control mice. (**a**) Absolute numbers of murine EpCAM, murine TROP2, and human EpCAM transcripts in 1 ng intestinal total RNA were determined using quantitative RT-PCR. Data represent means ± SD in samples from 4 mice in each group (*n* = 4). ** *p* < 0.01, **** *p* < 0.0001 using a one-way ANOVA with Tukey’s multiple comparison test. n.d. means not detected. (**b**) Immunofluorescence microscopy verifying expression of murine EpCAM (green) and murine TROP2 (red) in the small intestines of wild type (WT) and murine TROP2 rescue (T2R) mice. Nuclei were stained with 4′,6-diamidino-2-phenylindole (DAPI) (blue). Scale bars, 100 μm. (**c**) Immunofluorescence microscopy verifying expression of murine EpCAM (green) and human EpCAM (red) in the small intestines of WT and human EpCAM rescue (hEpR) mice. Nuclei were stained with DAPI (blue). Scale bars, 100 μm.

**Figure 2 cells-09-01847-f002:**
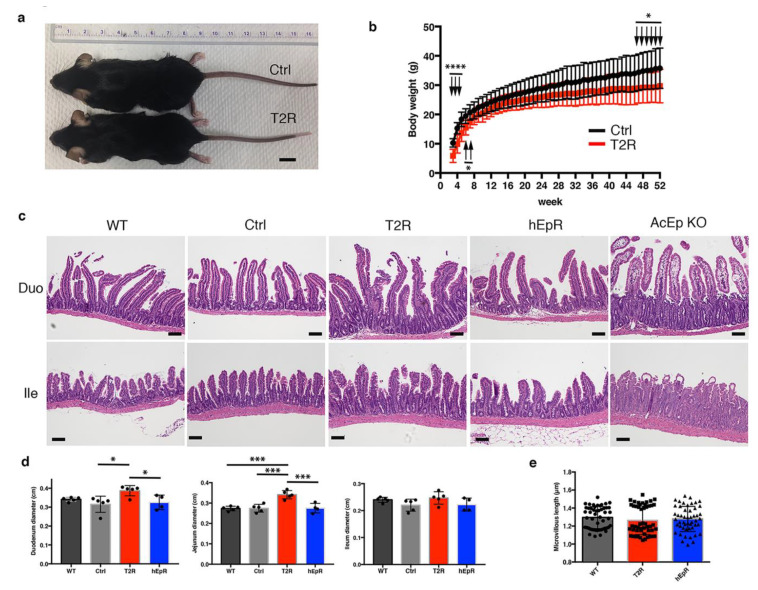
TROP2 expression in IEC prevents congenital tufting enteropathy in EpCAM knockout mice. (**a**) Representative photographs of littermate control mice (EpCAM^+/−^ TROP2^Tg^; Ctrl (T2)) and T2R (EpCAM^−/−^ TROP2^Tg^) mice at 8 weeks of age. Scale bar, 1 cm. (**b**) Serial body weights of littermate Ctrl and T2R mice over 1 year. Data depicted represent means ± SD for 32 Ctrl mice and 24 T2R mice at each time point. * *p* < 0.05, **** *p* < 0.0001 using a multiple t test and the Holm–Sidak method. (**c**) Representative H & E-stained histologic sections of small intestine from relevant mouse strains. AcEp KO designates tissue obtained from 8-week-old *EpCAM^fl/fl^ ROSA^CreER^* transgenic mice 7 days after initiation of tamoxifen treatment. Scale bars, 100 μm. (**d**) Intestinal diameters from relevant mice. Data represent means ± SD for 5 mice in WT, Ctrl, and T2R groups and for 4 mice in the hEpR group. * *p* < 0.05, *** *p* < 0.001 using a one-way ANOVA with Tukey’s multiple comparison test. (**e**) Intestinal microvillus length determined in transmission electron micrographs of tissue of indicated mice. Data depicted represent means ± SD of 50 microvilli for each genotype.

**Figure 3 cells-09-01847-f003:**
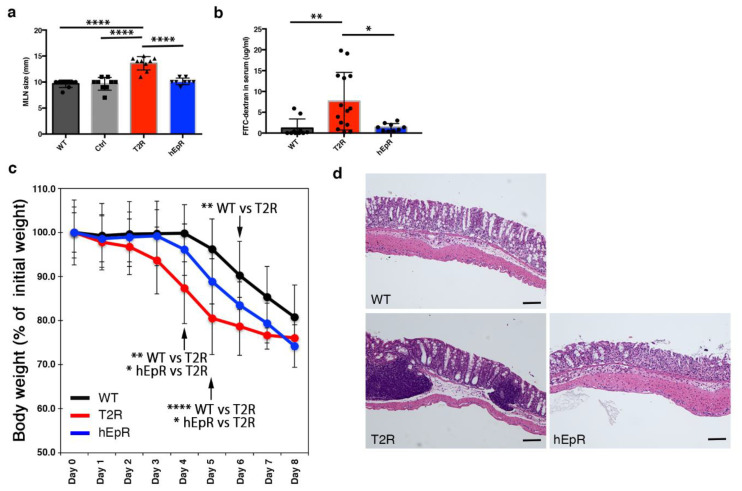
Abnormal intestinal barrier function in TROP2 rescue mice. (**a**) Sizes (maximal dimensions) of mesenteric lymph nodes from relevant mouse strains were determined. Data depicted represent means ± SD of dimensions obtained from 9 mice for each group (*n* = 9). **** *p* < 0.0001 as determined via one-way ANOVA with Tukey’s multiple comparison test. (**b**) FITC-dextran concentrations in sera from adult WT, T2R, and hEpR mice measured 4 h after administration of 4-kDa FITC-Dextran by gavage. * *p* < 0.05, ** *p* < 0.01 as determined via one-way ANOVA with Tukey’s multiple comparison test. (**c**) Serial body weights of adult female WT, T2R, and hEpR mice treated with 2.5% dextran sodium sulfate (DSS) in their drinking water. * *p* < 0.05, ** *p* < 0.01, **** *p* < 0.0001 as determined via one-way ANOVA with Tukey’s multiple comparison test. (**d**) Representative H & E-stained histologic sections of colon obtained on day 5 of DSS treatment. Scale bar, 100 μm.

**Figure 4 cells-09-01847-f004:**
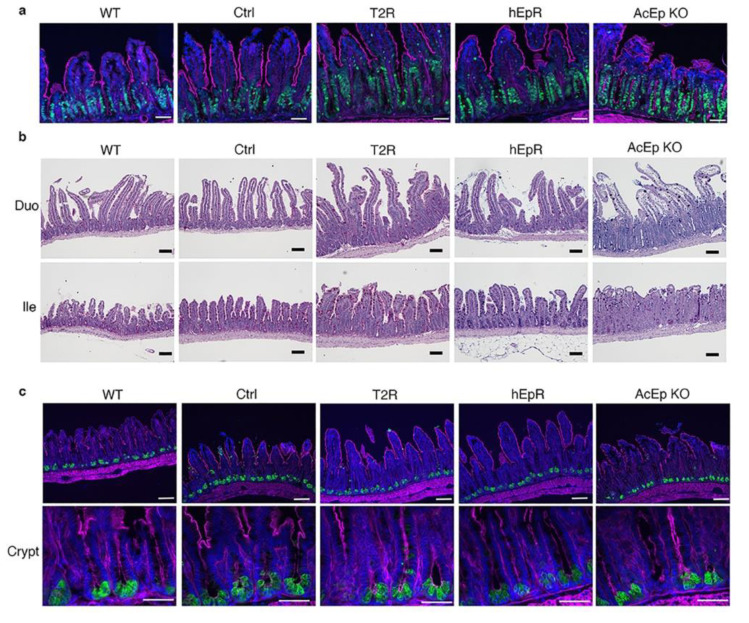
Characterization of IEC in TROP2 rescue mice and relevant controls. (**a**) Detection of proliferating cells in intestinal crypts by immunostaining. Ki-67 (green), F-actin (magenta), nuclei/DAPI (blue). Scale bars, 50 μm. (**b**) Goblet cells are visualized in fixed tissue sections using Alcian-PAS staining and light microscopy. Scale bar, 100 μm. (**c**) Identification of lysozyme-expressing Paneth cells in ileal tissue by immunostaining. Lysozyme (green), F-actin (magenta), and nuclei/DAPI (blue). Upper panel scale bars, 50 μm. Lower panel scale bars, 50 μm.

**Figure 5 cells-09-01847-f005:**
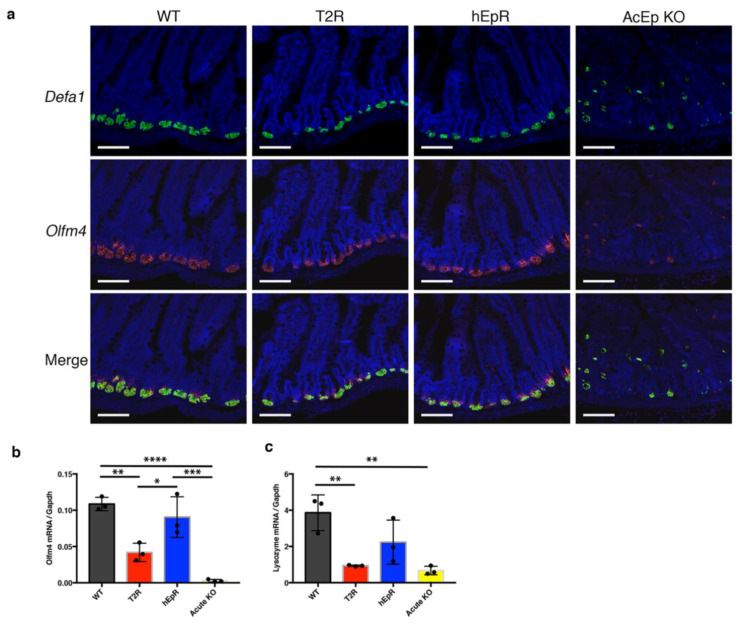
Diminished paneth cells and stem cells in TROP2 rescue mice. (**a**) Paneth cells (Defa1^+^, green) and stem cells (Olfm4^+^, red) were visualized in jejunum by RNA-Scope in situ hybridization and fluorescence microscopy. Scale bars, 100 μm. (**b**,**c**) qPCR analysis was performed to determine relative levels of mRNA expression in ileal tissue normalized to glyceraldehyde-3-phosphate dehydrogenase (GAPDH) mRNA expression. Values depicted represent means ± SE of samples from 3 different mice (*n* = 3). * *p* < 0.05, ** *p* < 0.01, *** *p* < 0.001, **** *p* < 0.0001 as determined using one-way ANOVA with Tukey’s multiple comparison test.

**Figure 6 cells-09-01847-f006:**
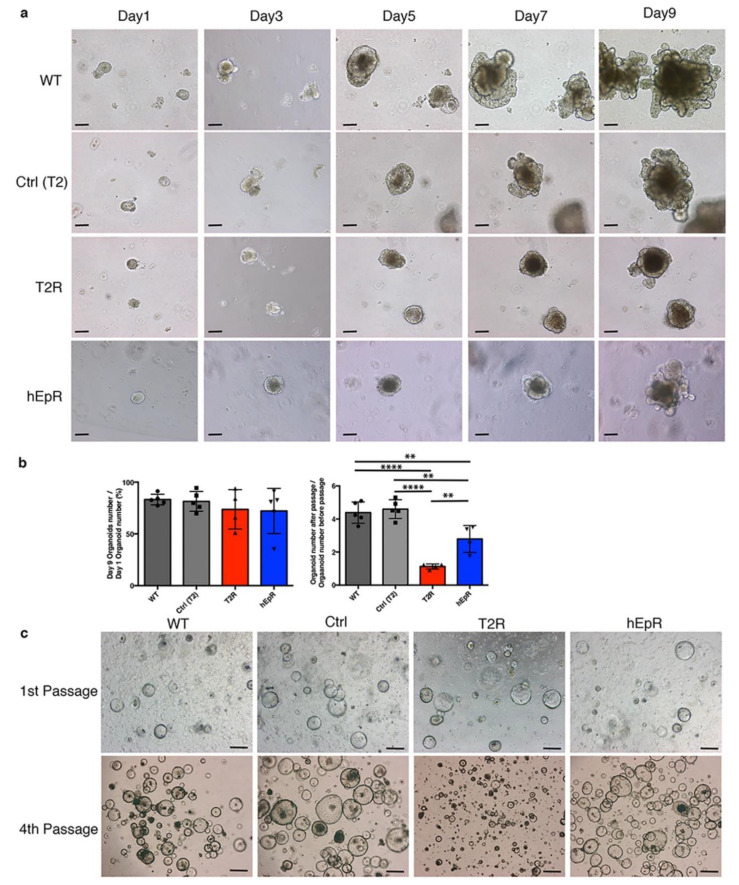
Characterization of organoids and spheroids from TROP2 rescue mice and relevant controls. (**a**) Representative serial phase contrast photomicrographs of IEC organoids obtained from WT, Ctrl (T2), T2R, and hEpR intestinal crypts in vitro. Scale bars, 100 μm. (**b**) Forming and passaging efficiencies of organoids from relevant mice. Organoid-forming efficiencies represent ratios of organoid numbers per well at day 9 to crypt numbers per well at day 1 (left panel). Aggregate data (means ± SD) from four independent experiments are presented. Ten wells were examined in each experiment. Organoid passage efficiencies represent ratios of organoid numbers/well at day 9 of secondary cultures to organoid numbers/well at day 9 of primary cultures (right panel). Aggregate data (means ± SD) from four independent experiments are presented. Six wells were examined in each experiment. ** *p* < 0.01, **** *p* < 0.0001 as determined via one-way ANOVA with Tukey’s multiple comparison test. (**c**) Representative phase contrast photomicrographs of spheroids from WT, Ctrl(T2), T2R, and hEpR mice on day 2 after initial passage and fourth passage as indicated. Scale bars, 100 μm.

**Figure 7 cells-09-01847-f007:**
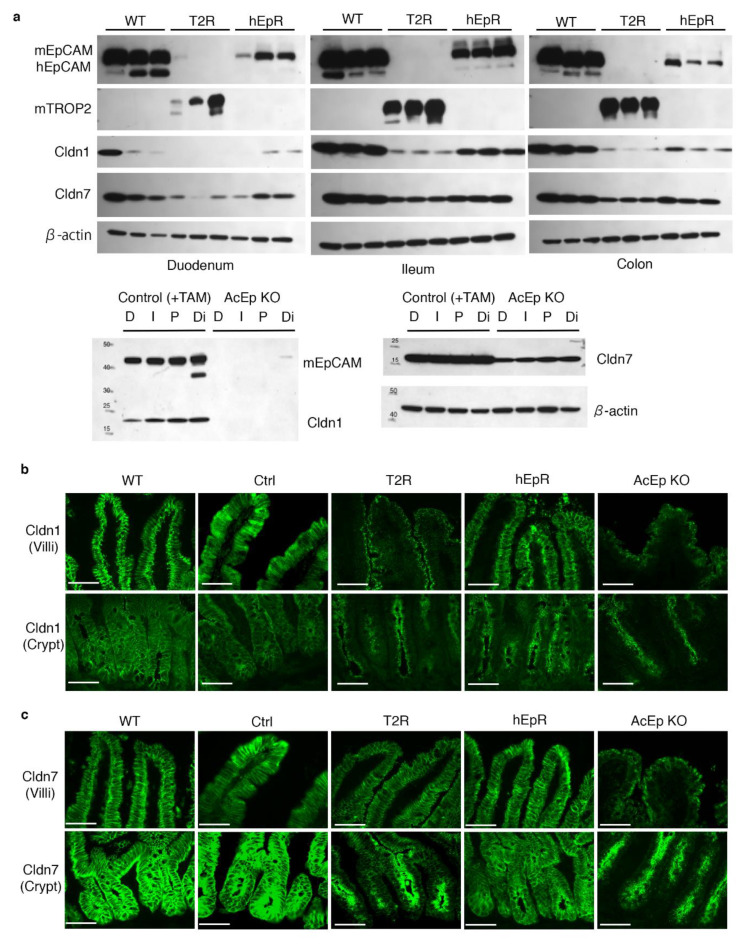
Decreased expression and mislocalization of claudin-1 and claudin-7 in TROP2 rescue mice. (**a**) Normalized amounts of intestinal tissue lysate proteins were resolved using SDS-PAGE and immunoblotted with anti-hEpCAM, anti-mEpCAM, anti-mTROP2, anti-claudin-1, anti-claudin-7, and anti-β-actin. Specific proteins were detected with HRP-labeled secondary Ab and chemiluminescence. D: Duodenum, I: Ileum, P: Proximal colon, Di: Distal colon, and TAM: tamoxifen treated. (**b**) Claudin-1 was detected and localized via immunostaining in ileum villi and crypts of relevant mice. Claudin-1, green. Scale bars, 50 μm. (**c**) Claudin-7 was detected and localized via immunostaining in ileum villi and crypts of relevant mice. Claudin-7, green. Scale bars, 50 μm.

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
