# Peer review of "Amelioration of Congenital Tufting Enteropathy in EpCAM (TROP1)-Deficient Mice via Heterotopic Expression of TROP2 in Intestinal Epithelial Cells"

_cells, 2020, doi:10.3390/cells9081847_

Round 1
Reviewer 1 Report
In figure 1 nuclear stain should be provided for all panels to ensure negative staining is obtained in an appropriate area.
Relevance for differences in growth between genders of T2R mice is not provided. Data regarding barrier dysfunction and DSS colitis should be investigated by gender to evaluate for functional differences which may accompany weight differences.
Regarding pathologic findings in the 2 models. Clearer quantitation of defects should be provided. Microvillus length (by EM) is provided. Crypt hyperplasia appears to be present in representative pathology of T2R and ACEpKO. Given villus atrophy and crypt hyperplasia in CTE patients, villus length/height and crypt height quantifications should be included. Additionally, ACEpKO mice have changes in the lamina propria of villi which should be shown at higher magnification and considered. Relevance for diameter of measurement and findings is not provided. This appears to be used as a surrogate measure of villus height, but diameter can be effected by more than just villus height.
Quantitation of histologic changes after DSS should be provided using a validated scale.
The authors show changes in Paneth cells in the novel model. Numbers of Paneth cells should be counted and compared. Decreases in Paneth cells have been reported in in patients with CTE and murine organoids. A rationale for their inquiry, relevance of the Paneth cell findings in context of human disease knowledge should be included.
T2R organoids are said to have fewer buds. Quantification of this finding should be provided.
Location/distribution of claudin 1 appears to be changed in T2R and ACEpKO mice/and Claudin 7 in AcEpKO. Improved resolution and higher magnification images should be included.
Author Response
Dear Reviewer,
Thank you very much for your valuable comments and helpful suggestions, which helped us improve the quality of our paper. Based on your suggestions, we have revised our manuscript. The experiments for this research were conducted in a laboratory of the NIH in Bethesda, MD, under the direction of Dr. Mark Udey. However, since his departure from the NIH in 2017, the laboratory has been closed. As a result, we have been unable to perform additional experiments. Nevertheless, we have exerted our efforts to address your comments to the extent possible. We have also updated the presentation of some of the figures in the revised manuscript. The modifications made in response to the all reviewers’ comments and suggestions are highlighted in yellow in the revised manuscript.
Attached please find our responses to the comments.
Thank you once again for offering us the opportunity to submit a revision and considering this paper for publication.
Sincerely on the behalf of the authors,
Gaku Nakato

Reviewer 2 Report
Report:
Nakato et al has submitted a manuscript entitled “Amelioration of congenital tufting enteropathy (CTE) in EpCAM (TROP1)-deficient mice via heterotopic expression of TROP2 in intestinal epithelial cells” for publication in Cells. The authors want to emphasize the importance of EpCAM or TROP2 in preventing CTE in EpCAM knockout mice. The TROP2 rescue mice were smaller than controls, while the EpCAM rescue mice were not. There were several abnormalities in phenotypic and functional characteristics in the TROP2 rescue mice compared to the wild type and EpCAM rescue mice. The authors concluded that EpCAM and TROP2 exhibited functional redundancy, but they are not equivalent in many ways.
The study is well designed. It is not surprising that TROP2 protein will not be able to compensate the function of EpCAM as they are different. However, the manuscript lacks several major discussions in their observations. Some of the statements are not justified with statistical analysis.
Major comments:
- The author should report the similarities between mEpCAM and hEpCAM as it’s not clear why transgenic mice were developed for hEpCAM instead of mEpCAM. Otherwise, it will be more ideal if you compare between hEpCAM and hTROP2 rather than comparing hEpCAM and mTROP2 transgenic mice. Please clarify.
- In both mTROP2 and hEpCAM transgenic mice, there was a 2-6-fold increased expression of either mTROP2 or hEpCAM transcripts compared to wild type mice. Please clarify whether the increased expression of those genes in those mice has any beneficial or detrimental effect compared to wildtype mice.
- The authors should discuss whether any gastrointestinal disorders like diarrhea have been detected in mTROP2 tg mice. The inflammatory markers have also not been discussed. That information is needed to order to better understand the differences in gastrointestinal symptoms between all groups of animals and in order to determine whether the loss of body weight is the cause of dehydration/diarrhea.
- In figure 2d, it says the data has been generated from 5 mice. However, the hEpR group has 4 data point instead of 5. Please clarify.
- Several statements like “proliferating cells were clearly increased in the ileum (line 335),” and “lysozyme+ Paneth cells were present in similar numbers and locations in crypts of mice (lines 340-341)” are not supported by quantification data. Please provide the quantitation data for those markers. If the T2R animals had more dextran FITC in the serum, they might have more leaky gut, which will also lead to the production of more lysozyme by Paneth cells to counteract those invading microorganisms. Please clarify why there is no increase in lysozyme + cells in the crypts and what that means in terms of leaky gut.
- Figure 4 legends do not match with the text discussed in lines 335-341.
- The authors have shown in figure 4c that there was no change in the lysozyme+ Paneth cell population. However in figure 5, the authors have shown the reduction of lysozyme mRNA/GAPDH again. The difference in lysozyme protein and gene expression has not been discussed properly. If the protein is important for protection, the mRNA values may not reflect the exact scenario in T2R mice. In that case, the statement saying diminished Paneth cells and stem cells in TROP2 rescue mice is not valid based on the mRNA data.
- In figure 6b, the authors have shown a significant reduction in organoid numbers after the passage in hEpR group compared to Ctrl. However, the representative picture in 6C does not represent the same story as depicted in figure 6b.
- Please clarify why in Figure 7a, 2 out of 3 samples from wildtype mice have low to none Cldn1 expression. Also, please define which part of small intestine has been stained for Figures 7b and 7c.
- Line 467-468: The authors have concluded a reduction in the frequency and/or function of Paneth cells in T2R mice that were detected in vivo and in vitro. The data does not support this conclusion. Moreover, no in vitro data has been presented to justify this finding.
Minor comment:
- Lines 487-489 in discussion section are redundant with the lines 66-68 in introduction section.
- Line 508: typographical error for the spelling expression.
Author Response

(The authors gave the same response as above.)

Reviewer 3 Report
To more understand functions of TROP1 and TROP2 that are homologus cell surface protein in the intestinal epithelium, the authors made full use of T2R and hEpR mice, respectively. The authors conclude that TROP1 and TROP2 exhibit functional redundancy, but they are not equivalent. The manuscript generates fruitful information for the function of the proteins. I would request to rewrite the discussion in more detail for readers. Especially, the relationship among claudins, TROP1, and TROP2 is poor discussion.
Minor comments
1) Page3,line84: Please show the approved number for animal experiments.
2) Page5,line158: Please add the sentence shown in "Electron Microscopy". Details of protocols are described in the supplementary information.
3) Page6,line197, Page15,line438 and Page17,line471: 293 cells to HEK 293 cells.
4) Page8,line260: Why were only male T2R mice smaller than corresponding controls, but not female T2R mice? Please describe the reason in discussion.
5) Page11-12: The authors show that Ki-67 positive cells are increased in the proliferating zone of crypt-villus axis in the T2R intestine. However, intestinal stem cells (ISCs) are decreased in the intestine. If the rate is maintained, ISCs are exhausted. Please explain the contradiction in discussion.
6) Page12,line370-371: It is hard to understand the author's conclusion. Please rewrite the sentences.
7) Page17 and 18: In discussion, there are space.
8) Page17,line479-486: It is very hard to understand between claudin metabolism and stem cell abnormalities. What is claudin metabolism? And, What is stem cell abnormalities? Proliferation and/or differentiation, or maintenance? Furthermore, what is physical location of Paneth and stem cells ? It means cell-to cell interaction?
9) Page17,line487~: Suddenly, HAI-2 (SPINT2) is appeared. Please explain a functional pathway that links the gene or protein.
10) Page18,line490: Probably, "as" is deleted. Please add it.
11) Page18,line498-500: Paneth cells is important for making ISC niche. The cells secrete a variety of signaling molecules including Wnts. When Wnt3a (for example) is added in the spheroids, the treated spheroids are rescules? It is difficult to understand Paneth cell abnormalities. What kind of abnormalities?
12) Figure S1: What is the data in middle? Total of male and female body weight?
Author Response

(The authors gave the same response as above.)

Round 2
Reviewer 1 Report
Overall the authors were not able to address most of the concerns raised in the review due to lab closure and thus lack of experimental material. That being said, overall the conclusions, methods and results are sound.
Minor:
The authors should make effort to include the rationale and references regarding gender differences and paneth cell related findings (which were provided in the rebuttal) to the discussion.
Interpretation of Claudin-1 findings in 7B should be more clearly interpreted and included.
Author Response
Dear Reviewer,
Thank you very much for your comments.
The modifications made in response to the comments are highlighted in yellow in the additionally revised manuscript.
Attached please find our responses to the comments.
We appreciate the time and effort that you devoted to review of our manuscript.
Sincerely and on the behalf of the authors,
Gaku Nakato

Reviewer 2 Report
The authors have addressed the critiques properly. The paper is now good for publication.
Author Response
Dear Reviewer,
Thank you very much for your comments.
We appreciate the time and effort that you devoted to review of our revised manuscript.
Sincerely and on the behalf of the authors,
Gaku Nakato